# Detecting Human Actions in Drone Images Using YoloV5 and Stochastic Gradient Boosting

**DOI:** 10.3390/s22187020

**Published:** 2022-09-16

**Authors:** Tasweer Ahmad, Marc Cavazza, Yutaka Matsuo, Helmut Prendinger

**Affiliations:** 1Department of Electrical and Computer Engineering, COMSATS University Islamabad, Islamabad 45550, Pakistan; 2National Institute of Informatics, Tokyo 101-8430, Japan; 3Department of Engineering, The University of Tokyo, Tokyo 113-8654, Japan

**Keywords:** action detection, YoloV5, gradient boosting classifier

## Abstract

Human action recognition and detection from unmanned aerial vehicles (UAVs), or drones, has emerged as a popular technical challenge in recent years, since it is related to many use case scenarios from environmental monitoring to search and rescue. It faces a number of difficulties mainly due to image acquisition and contents, and processing constraints. Since drones’ flying conditions constrain image acquisition, human subjects may appear in images at variable scales, orientations, and occlusion, which makes action recognition more difficult. We explore low-resource methods for ML (machine learning)-based action recognition using a previously collected real-world dataset (the “Okutama-Action” dataset). This dataset contains representative situations for action recognition, yet is controlled for image acquisition parameters such as camera angle or flight altitude. We investigate a combination of object recognition and classifier techniques to support single-image action identification. Our architecture integrates YoloV5 with a gradient boosting classifier; the rationale is to use a scalable and efficient object recognition system coupled with a classifier that is able to incorporate samples of variable difficulty. In an ablation study, we test different architectures of YoloV5 and evaluate the performance of our method on Okutama-Action dataset. Our approach outperformed previous architectures applied to the Okutama dataset, which differed by their object identification and classification pipeline: we hypothesize that this is a consequence of both YoloV5 performance and the overall adequacy of our pipeline to the specificities of the Okutama dataset in terms of bias–variance tradeoff.

## 1. Introduction

In recent years, drones and unmanned aerial vehicles (UAVs) have found numerous applications in urban surveillance, search and rescue, and situational awareness applications. Several of these applications require the ability to recognize actions from UAV cameras, either through video or single-image analysis. Human action recognition is considered to be a challenging task, which has been fairly addressed over the last decade [1]. However, extending this task to drone-captured images and videos is an emerging topic. Human action recognition is a well-studied problem which is categorized into (i) pose-based [2,3,4], (ii) single-image-based [5,6,7], and (iii) video-based action recognition [8,9,10]. However, detecting actions in single images is a less explored area because it faces the problem of the unavailability of annotated temporal data for action detection [11,12,13]. This task requires the integration of components for entity detection, and classification that can be adapted to the distribution of target situations, as well as practical deployment constraints. With the availability of popular and efficient object detection methods such as Yolo, it becomes possible to envision solutions that incorporate a detection module on top of object recognition. Eweiwi et al. [2] built an efficient pose-based action recognition using histograms of relative location, velocity, etc., and thus learned a compact and discriminative representation. Wang et al. [3] recognized human actions by obtaining K-best estimations and additional segmentation cues. An encoder-based approach was proposed by [4], which efficiently encodes 3D skeleton information using pose descriptor and uses extreme learning machine for classification. A single-image-based action recognition was devised by [5], which segments out the human body into five parts: head, torso, arms, hands, and legs. A semi-FCN network detects each of these five body parts and then Action ResNet predicts the action for each body parts. Finally, an SVM fuses these five body part actions and thus recognizes the entire body action. Sreela et al. [6] evolved the action recognition model using residual neural network as feature extractor and support vector machine as the classifier. For single-image action recognition, Liu et al. [7] suppressed misleading context of person bounding boxes using guided-loss activation with ResNet-50 deep learning architecture. Several video-based action recognition techniques were summarized by [8] for three types of benchmark datasets, i.e., single-viewpoint, multi-viewpoint, and RGB-depth video. Pareek et al. [9] briefly covered the human action recognition techniques using machine learning and deep learning methods for the years 2011–2019. Pham et al. [10] presented most important deep learning models for action recognition, analyzed their performance, and then discussed future prospects and challenges for recognizing actions in realistic videos. Rohrbach et al. [11] prepared a dataset for kitchen activities. Singh et al. [12] used LSTM for building a multi-stream bidirectional network for action detection. Reinforcement learning was used by [13] for detecting different actions in videos.

In previous work on pedestrian detection, Zhang et al. [14] showed the effectiveness of gradient boosting. Such a technique can also be applied to action detection and recognition from single images in UAV data. Therefore, we adopted gradient boosting on top of Yolo-based pedestrian detection [14] as an action classifier.

Human action detection in aerial images is closely related to pedestrian detection and tracking in UAV videos [14,15]. Liu et al. [16] applied the Yolo architecture to small object detection in UAV image data. Mittal et al. [17] highlighted various small object detection techniques for UAV video data. Shinde et al. [18] employed a vanilla Yolo architecture for detection and localization of human activities in the Liris dataset [19]. Very recently, Jung et al. [20] improved the YoloV5 architecture by modifying convolution layer architecture and activation function for efficient object and person detection in aerial images of the VisDrone dataset. Caputo et al. [21] examined lightweight versions such as YoloV5s and YoloV5m for speedy search and rescue of humans in dangerous situations. The authors evaluated their models on the two newly collected HERIDAL and SAR datasets.

The Yolo series [22,23,24,25] has played an important role as a single-stage detector in object detection and action detection tasks. In this paper, we specifically investigate YoloV5 with gradient boosting for solving the problem of human action recognition and localization in drone videos. Our proposed method is able to address the challenges of drone camera motion, smaller actor size, and the limited size of aerial action datasets for training the model.

The main contribution of this paper is to suggest an action recognition pipeline compatible with use in UAVs on real-world datasets, such as the Okutama-Action dataset, based on state-of-the-art object recognition techniques. As a secondary contribution, we also experiment with different variants of YoloV5 and propose a compromise between model performance, size, and computational complexity.

In the remainder of this paper, Section 2 summarizes the related work. We explain our proposed methodology in Section 3, while experimental details and results are discussed in Section 4. Finally, we conclude our paper in Section 5.

## 2. Related Work

Over the last decade, recognizing human actions in videos has been challenging and an active area of research among the computer vision community [26,27,28,29,30,31,32,33,34]. However, the recognition and detection of actions in aerial images is a less developed area, and differs from previous work that simply adopts the perspective of pedestrians in the scene [31,35].

Over the course of time, several aerial action datasets have been collected. The UCF-ARG dataset [36] contains 10 realistic human actions in the three settings of ground, rooftop, and aerial triplets. This is considered to be a challenging dataset as the dataset contains various instances of camera motion and humans tend to occupy only a few pixels within images. Perera et al. [37] recorded a slow and low-altitude (about 10 ft) UAV video dataset for detecting 13 different gestures in aerial videos. These gestures are mainly related to UAV navigation and aircraft handling. The authors investigated a pose-based convolutional neural network (CNN) for this work. Ding et al. [38] overcame the challenging problem of heavy computations by devising a lightweight model for real-world drone action recognition. The backbone architecture for this method contains a temporal segment network with MobileNetV3, where temporal structures are responsible for capturing self-attention and the focal loss emphasizes misclassified samples. Geraldes et al. [39] proposed a UAV-based situational awareness system called Person-Action-Locator (PAL). The PAL system is robust enough to automatically detect people and then recognize their actions in near-real-time.

Mliki et al. [40] introduced a two-stage methodology for recognizing human activities in UAV-captured videos. The first stage is responsible for generating human/non-human entities and human activity models using CNN. The second inference phase employs CNN-based human activity modeling to recognize human activities by using majority voting for the whole video sequence. Choi et al. [41] investigated the emerging problem of action recognition in drone videos using unsupervised and semi-supervised domain adaptation. The proposed method transfers the knowledge from source to target domain using video and an instance-based adaptation methodology. The authors also created a dataset of 5250 videos for evaluating their proposed method.

Barekatain et al. [42] presented the Okutama-Action dataset as a concurrent aerial view dataset for human action recognition and detection. This dataset contains 43 min of video with 12 different action categories. The Okutama-Action dataset poses a generic challenge due to the realistic condition of video acquisition that results in dynamic transition of actions, and challenges specific to single-image action detection, such as significant changes in scale and aspect ratio of the subject, abrupt camera movement, and side and top views of the subjects, as well as multi-labeled actors.

If we could devise a working pipeline for such a dataset, it will increase its suitability to process real-world situations.

## 3. Methodology

The offline pipeline of our proposed method for action detection and recognition in drone images is illustrated in Figure 1. In the first stage, a camera mounted either at 45${}^{\circ }$ or 90${}^{\circ }$ on a drone captures the outdoor scene. In the second stage, these drone-captured images are input into an anchor-free single-staged YoloV5 detector. A gradient boosting classifier accepts the output of the Yolo detector and detects and recognizes different actions. The final stage draws the bounding boxes and confidence score for each prediction. Moreover, we present a detailed diagram of YoloV5 architecture and gradient boosting classifier in Figure 2.

### 3.1. YoloV5 for Drone Action Detection

Two-stage object detectors have been a popular choice among the research community and include R-CNN series [43,44,45]. As compared to two-stage detectors, single-stage detectors are faster because they can simultaneously predict the bounding box and the class of objects. However, there is a compromise for the slight drop of accuracy for single-stage object detectors. The prominent single-stage detector may contain Yolo series [22,23,24,25], SSD [46], and RetinaNet [47].

YoloV5 is one of the most famous detection algorithms due to its fast speed and high accuracy. YoloV5 divides the images into a grid system, where each cell in the grid is responsible for detecting objects within itself. This approach provides a specific advantage when multiple objects are involved, which is of particular importance to multi-person action recognition [48]. YoloV5 is the most recent model of the Yolo detection family and it contains the best architectures among the Yolo family. We interchangeably use the object detection and action detection terminology.

In a broader sense, YoloV5 comprises three main architectures: (i) backbone architecture, (ii) neck layer of feature pyramids, and (iii) bottleneck layer with prediction heads for action detection. The backbone architecture generally contains VGG [49], ResNet [50], DenseNet [51], MobileNet [52], EfficientNet [53], and CSPDarknet53. PANet aggregates features in the neck layer which includes several bottom-up paths and several top-down paths for upsampling and downsampling. The path-aggregation block generally consists of FPN [54], PANet [55], NAS-FPN [56], Bi-FPN [57], ASFF [58], and SFAM [59]. In the pipeline of work, the bottleneck layer of prediction heads may comprise the one-stage or two-stage detector.

There are five different models for YoloV5: YoloV5s, YoloV5m, YoloV5l, YoloV5x, and YoloV5n, which offer various options adapted to different computational and deployment constraints. We choose our baseline as YoloV5 which includes CSPDarknet53 as backbone, PANet as neck, and Yolo detection as head layer for a single-stage detector [22]. While training, we noticed that YoloV5x outperforms other models, i.e., YoloV5s, YoloV5m, YoloV5l, and YoloV5n. One clear disadvantage of YoloV5x is longer training time and larger model size. YoloV5n is the tiny version of YoloV5, which reduces one-third of the depth of YoloV5s and, therefore, results in 75% reduction in model parameters (7.5 M to 1.9 M), which make it an ideal choice for deploying on mobile devices and CPU-only machines. In YoloV5 architecture, there is other recent advancement, such as YoloV5-P5 and YoloV5-P6. YoloV5-P5 models have three output layers, P3, P4, and P5, with stride sizes of 8, 16, and 32 at an image size of 640 × 640. On the other hand, YoloV5-P6 models have four output layers, P3, P4, P5, and P6, having stride sizes of 8, 16, 32, and 64, which were trained for image size of 1280 × 1280. YoloV5-P6 with stride size of 64 works well for detecting larger objects in high-resolution training images. By the time of study, YoloV7 was not released; however, YoloV5 as a single-stage detector is an appropriate choice for single-image action detection, since it brings different variants of YoloV5, thus offering a compromise between usability and performance for a UAV scenario.

### 3.2. Gradient Boosting Classifier

Previous work on the Okutama dataset explored various architectures based on a pipeline of entity recognition and temporal feature recognition using specialized variants of a classical CNN–LSTM approach. Barekatain et al. [42], more specifically, used SSD-classifer and ensembled two-streams as RGB and optical flow for action detection where both streams work in a complimentary fashion. Geraldes et al. [39] also ensembled the output of two architectures, POINet and ActivityNet. POINet detects the bounding box for each person in the scene using CNN; meanwhile, ActivityNet employs LSTM and computes the temporal features and action labels for each person.

The new architecture we introduce here adopts a similar philosophy, but aims at upgrading individual components for object identification and action classification, while also substituting boosting into the RNN component. While boosting is equally able to deal with temporal information, this might be a smaller issue with image-based action recognition. In addition, it offers more flexibility in terms of learning behavior, and is gaining popularity for action recognition tasks.

It should be noted that some previous work has explored a tight integration between CNN and boosting for vision tasks. However, by incorporating boosting weights into deep learning architecture [60], our use of boosting follows the previous pipeline processing philosophy with independent processing stages.

We have the same rationale for our work by using gradient boosting, which ensembles the output of different classifiers working in a complementary fashion. The Okutama-Action dataset contains images of camera angles with 45and 90, at altitude of 30 m and varying the distances between subjects and camera. The dataset images are also self-occluded and occluded by different objects in the scene. All these factors contribute to high variance for the Okutama-Action dataset.

Gradient boosting (GB) classifiers are good at mitigating high variance and high bias, which may cause overfitting and underfitting problems, respectively. Gradient boosting significantly reduces the high variance problem by decreasing the learning rate, because a higher learning rate more aggressively captures the variation among training samples [61]. On the other hand, gradient boosting controls high bias by increasing the boosting rounds, in which each round corresponds to the addition of a new decision tree [62,63]. The bias term consistently decreases as the number of boosting rounds is increased. Gradient boosting builds a mechanism for reducing the bias and variance in expected prediction error. Using gradient boosting, when a model is trained with low learning rate and higher number of boosting rounds, results in low bias and variance and correspondingly improves the model performance. This is another strong motivation for using gradient boosting as a powerful general-purpose learning algorithm in our work.

Gradient boosting is conceived to be better than other machine learning algorithms, such as bagging and random forest decision trees, because it involves weight adjustment using decision tree predictions. GB also involves cross-validation, efficient handling of missing data, regularization to avoid overfitting, tree pruning, and paralyzed tree building, [64]. Gradient boosting fits the nonlinear (piecewise linear) decision boundary, while SVM always fits the linear boundaries even if the dataset is not linearly separable; therefore, GB brings more flexibility and, therefore, performs better than polynomial-based SVM approaches [65].

GB assigns different weights to different samples in such a manner that difficult-to-classify samples are weighted more whilst easily-classified samples receive less weight. Using gradient boosting, weak learners are sequentially added up to better classify the difficult samples. We employ log likelihood as loss function for the gradient boosting classifier. The gradient boosting is explained in Algorithm 1. The same concept of gradient boosting is also explained in Figure 3. During experimentation, we used 100 trees (boosting rounds) of maximum depth of 3 and a learning rate of 0.001. In addition, we used the Scikit-learn package for implementation of the gradient boosting algorithm [66].
**Algorithm 1** Gradient Boosting Algorithm**Inputs**: (i) $\left\{x_{i},y_{i}\right\}$, (ii) loss function, L(y,F(x)), (iii) No. of trees *M***Procedure**:(1) Initialize Model with Constant Value
(1)$$F_{0}\left(x\right)=argmin_{\gamma }\left(\sum_{i=1}^{n}L\left(y,\gamma \right)\right)$$(2) Iterate m=1 to *M*  (i) Compute Pseudo-residuals
(2)$$\gamma_{im}={\left[\frac{\partial L(y_{i}F\left(x_{i}\right))}{\partial F(x_{i})}\right]}_{F\left(x\right)=F_{m-1}\left(x\right)}^{i=1,...,n}$$  (ii) Fit a Base-Learner $h_{m}\left(x\right)$, input $\left\{\left(x_{i},\gamma_{im}\right)\right\}$  (iii)
(3)$$\gamma_{m}=argmin_{\gamma }\sum_{i=1}^{n}L\left(y_{i},F_{m-1}\left(x_{i}\right)+fl\right)$$  (iv) Update Model,
(4)$$F_{m}\left(x\right)=F_{m-1}\left(x\right)+\gamma_{m}h_{m}\left(x\right)$$

### 3.3. Description of Architecture

In our work, gradient boosting is implemented by increasing the frequency of difficult samples in order to better learn from these samples. Gradient boosting adjusts the weights based on the previous decision tree predictions. The residual error is computed and added to the initial values and then fine-tuned, so that the final prediction approaches closer to the ground-truth values.

The main challenges that occur during action detection in the Okutama-Action dataset are (i) human subjects are not well-exposed to the camera, as are subjects in images taken on the ground, (ii) subjects are of very small size, as compared to the whole image, and (iii) subjects occur from different camera viewpoints.

#### 3.3.1. Input

Following the common practice of object detection, the input image is expected to have a size of H×W×C, where *H* and *W* correspond to image height and width, whereas *C* represents the number of channels. In our case, the number of channels is set to three (RGB).

#### 3.3.2. Backbone

Multi-scale features are extracted using either ResNet [50], ResNeXt [67], or DenseNet [51] as encoder. These features are made compatible and input into a feature pyramid network (FPN). Feature map extraction at different stages 1∼N is represented correspondingly as $C_{1}∼C_{N}$.

#### 3.3.3. Feature Pyramid Network (FPN)

The feature pyramid network helps to detect objects at different scales. The lower-level layers in FPN have higher resolution with fewer semantic details, whilst higher-level layers have lower resolution with stronger semantic meaning. The residual connections fuse the features between different layers and thus facilitate smaller objects’ detection.

#### 3.3.4. Detection Heads

The prediction heads carry out per-pixel prediction, where a prediction is output from three heads of similar architectures, i.e., a 2D convolution → a group normalization → a rectified linear unit (ReLU). The three outputs of these heads are centerness head, class prediction head, and box-regression head, as shown in Figure 2.

#### 3.3.5. Network Settings

In our experiments, we used CSPDarknet53 and ResNet-50 as backbone architectures. We empirically choose all the network parameters, e.g., initial base learning rate was set to 0.001 with a weight decay of 0.0001 and a momentum of 0.9. The image mosaic parameter was set to high to take advantage of data augmentation. We ran our experiments for 200 epochs with batch size of 32. We implemented our method in the Pytorch platform to run these experiments [68], while the Scikit-learn library was used for implementing gradient boosting algorithm with 100 trees and depth-level of 3. Tesla P100-PCIE with cuda-10.2 was the hardware machine for this implementation.

## 4. Experiments

### 4.1. Dataset Description

The Okutama-Action dataset [42] consists of a 43-minute-long annotated video sequence of 12 different outdoor action categories with about 77,000 image frames. This is a challenging dataset because it includes abrupt camera motion, dynamic transition of actions, scale variation due to near–far movement of drone, and variation in aspect ratio of actor while performing different actions. Some example images of the Okutama-Action dataset are shown in Figure 4. Okutama-Action videos were captured using a DJI Phantom 4 UAV at a baseball field in Okutama, Japan. These twelve actions of the Okutama-Action dataset are grouped into three types:Human-to-human interaction (handshaking, hugging).Human-to-object interaction (reading, drinking, pushing/pulling, carrying, calling).No interaction (running, walking, lying, sitting, standing).

For capturing the action videos, the UAV was operated at altitude range of 30 m to 45 m and the gimbal angle was set as either 45° or 90°.

The Okutama-Action dataset covers two different scenarios for data collection in the morning and noon settings for incorporating different lighting conditions (sunny and cloudy). Additionally, this dataset was captured using two different drones operated by two different pilots with different speeds and maneuvers. For some of the videos, metadata was provided for altitude, speed, and gimbal angle. During data collection, a 4K high-resolution UAV-mounted camera was operating at 30 FPS.

### 4.2. Ablation Study

We evaluate the performance of our proposed models using precision, mAP@50%IoU, recall, and F1-score.

#### 4.2.1. Baseline Model

We define YoloV5s and YoloV5s6 as our baseline models. YoloV5s is termed as YoloV5-P5 and has three output layers with stride sizes of 8, 16, and 32 for image size 640 × 640, whilst YoloV5s6 is termed as YoloV5-P6, having four output layers of stride 8, 16, 32, and 64 with image size of 1280 × 1280. We report the performance using YoloV5s and YoloV5s6 in Table 1, whilst a performance chart for each category of actions, primary and secondary actions, is also reported in Table 2. Primary actions are performed by the single subject without any interaction with other subject or object (e.g., run, walk, lying, sit, stand). Meanwhile, secondary actions may involve interaction with some object (e.g., read, call, drink, actions require book, phone, bottle) or interaction with some other subject (e.g., handshake, hug).

#### 4.2.2. Comparison among YoloV5 Architectures

We conducted our experiment with different YoloV5 architectures and report our research findings in Table 3. We exercised different architectures of YoloV5-P5, such as YoloV5s, YoloV5n, YoloV5m, YoloV5l, and YoloV5x, and YoloV5-P6, such as YoloV5s6, YoloV5n6, YoloV5m6, YoloV5l6, and YoloV5x6. YoloV5n and YoloV5n6 are the smallest in size with a slight compromise in accuracy which makes them an ideal choice for deploying in drone applications where computational resources are always a constraint. Mainly, this experiment works offline but devises small sizes of Yolo architectures to work online, as shown in Table 3. The YoloV5-P6/64 output layer performs well for detecting larger objects in high-resolution images. YoloV5x and YoloV5x6 architectures are the largest in size and performed better than other architectures.

### 4.3. Comparison with State-of-the-Art Methods

We present a comparison of our proposed method with other state-of-the-art methods in Table 4. Barekatain et al. [42] used SSD for detecting actions in Okutama-Action dataset with image size of 512 × 512. The authors ran their experiments on RGB and optical flow streams and then combined the results of both streams for better accuracy. During experimentation, it was realized that the SSD model performed best when the camera angle was 45 degrees. It was also noticed that strongly related temporal actions, e.g., the *running* action, resulted in lower recognition accuracy because SSD sequentially performed detection in a frame-by-frame manner. Soleimani et al. [69] proposed a two-stage architecture for identifying the action categories in the Okutama-Action dataset. The first stage relies on SSD for finding objects of interest, whereas the second stage uses another CNN to learn the latent sub-space for associating aerial imagery and action labels. The main difference between SSD and Yolo architectures lies in the handling of bounding boxes, as SSD treats each bounding box prediction as a regression problem. Yolo architecture, on the other hand, computes non-maximum suppression and thus retains the final bounding box. Yolo methods are slightly better than SSD in detecting smaller objects. This is exactly the situation in our scenario due to the distance between flying drones and human subjects on the ground. This proposed architecture resulted in significant increase of 28.3% in mAP for 50% IoU. Geraldes et al. [39] devised POINet (Position and Orientation Invariant Neural Networks) by employing MobilenetV2 [70] and reported the action detection performance separately for primary and secondary actions. The performance of our proposed method in terms of primary and secondary actions is also better than [39]. In Table 4, we explicitly mentioned the type of classifier for each methodology and realized that the gradient boosting classifier with Yolo action detector shows better performance than the fully-connected feed-forward classifier with detectors. The qualitative detection results of our proposed methodology are presented in Figure 5.

## 5. Conclusions

Deep learning is a method of choice for drone action recognition and detection due to its high performance and easy deployment. In this paper, we propose a Yolo-based framework with gradient boosting for detecting and recognizing twelve different actions. We investigated the performance of our model on the Okutama-Action dataset, where it outperformed other methods for individual actions, primary and secondary actions. We evaluated the performance of our method for different settings, such as morning or noon, and varying gimbal angle, of 45 degree or 90 degree, for an altitude of 30 m. We also evaluated the performance of different variants of YoloV5. Since YoloV5 is efficient for detecting multiple objects in an image, and the Okutama-Action dataset contains concurrent actions in a frame, our proposed method achieved better results for single-image action detection than other methods in the literature that may include explicit temporal information, e.g., LSTM or 3D-CNN.

The main limitation of our work is that the performance of our algorithm may degrade if the gimbal angle becomes closer to 90° or if the drone flies above 30 m. Moreover, the speed of the drone may also negatively affect the performance of our algorithm due to motion blurring. In addition, low lighting conditions of early morning–evening and extreme weather conditions of raining, clouds, or snowing may severely degrade the performance of our method.

YoloV7 shows good results for pose estimation and is beneficial for single-image object detection; therefore, in future work, it is possible that YoloV7 can be a good choice for action detection. As an alternative to YoloV7, self-attention transformers and their variants, e.g., ViT or ScaleViT, are good candidates that could replace the pipeline of YoloV5 and gradient boosting for action detection.

## Figures and Tables

**Figure 1 sensors-22-07020-f001:**
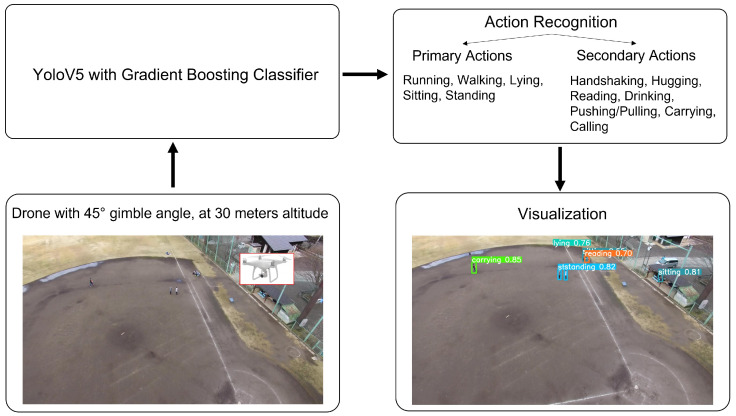
Overview of pipeline of work. First, the flying drone with a camera captures the images of the scene, which are input into the YoloV5 detector in an offline process. The gradient boosting classifier makes weight adjustments for difficult samples and re-trains the model for final action detection.

**Figure 2 sensors-22-07020-f002:**
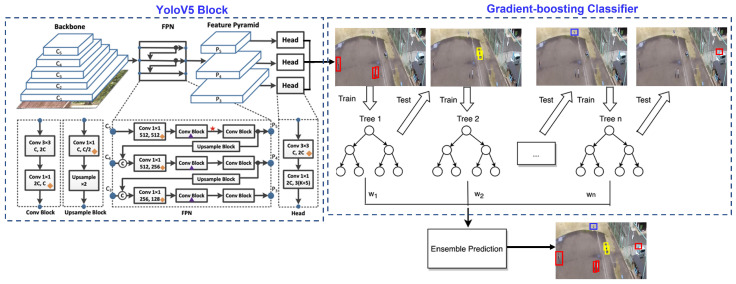
The left side of this figure explains the anchor-box free single-stage YoloV5 detector. YoloV5 architecture comprises of backbone architecture, neck layer as feature pyramid network (FPN), and prediction heads as bottleneck layers. The right side of this figure illustrates gradient boosting, which ensembles the predictions of the Yolo classifier for final action detection.

**Figure 3 sensors-22-07020-f003:**
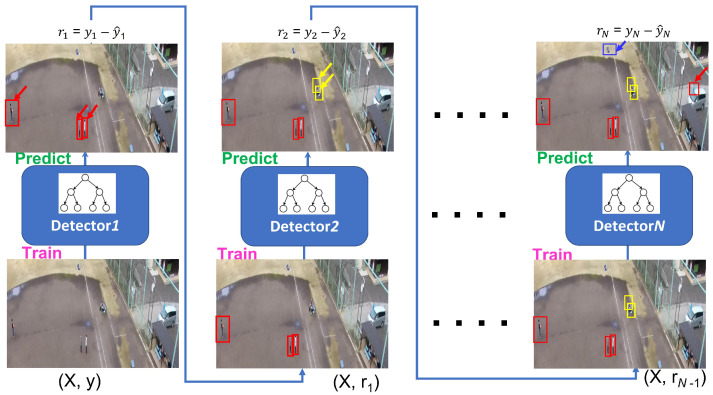
This diagram explains the gradient boosting classifier for learning and retraining on difficult instances. The boosting technique involves weight adjustment based on the previous decision tree prediction. The output features of YoloDector are encoded as feature vectors for input into the boosting classifier. The regularization is introduced for this model using shallow boosting trees.

**Figure 4 sensors-22-07020-f004:**
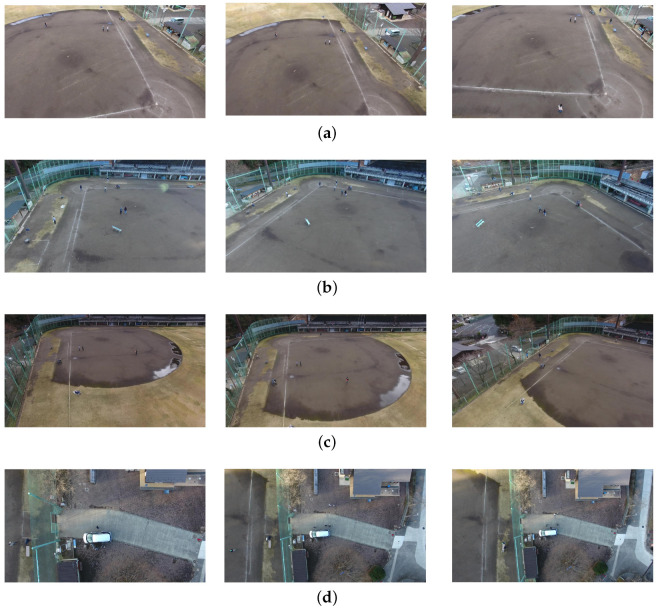
Some sample images from the Okutama-Action dataset. (**a**) Drone1, morning. Frame 1, 225, 2220 in video 1.1.1; (**b**) Drone1, noon. Frame 2, 270, 685 in video 1.2.2; (**c**) Drone2, morning. Frame 1, 200, 870 in video 2.1.1; (**d**) Drone2, noon. Frame 175, 325, 700 in video 2.2.5. In this setting, camera tilt is near to 90 degrees.

**Figure 5 sensors-22-07020-f005:**
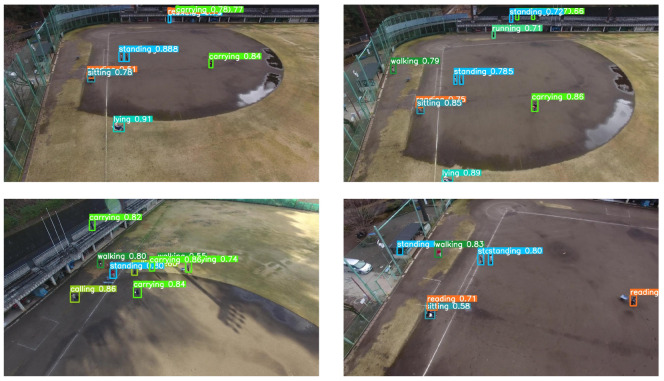
Inference results for action detection in drone images. Each color bounding box corresponds to a separate action category.

**Table 1 sensors-22-07020-t001:** Comparison among YoloV5s-P5 and YoloV5s6-P6 as baseline architectures. mAP is computed in (%) at IoU = 0.5:0.05:0.95, while mAP@0.5 is computed at 50% IoU.

Method	mAP	mAP@0.5	Recall	F-Score
YoloV5s	64.6	62.9	65.8	65.2
YoloV5s6	68.7	63.4	61.4	64.8

**Table 2 sensors-22-07020-t002:** Performance evaluation for each individual action category for YoloV5s6 architecture.

	Method	mAP	mAP@0.5	Recall	F-Score
Primary Actions	run	50.9	46.6	49.4	50.1
	walk	67.8	69.8	70.5	69.1
	lying	69.8	83.1	79.7	74.4
	sit	70.2	66.4	67.4	68.8
	stand	67.8	67.6	69.4	68.6
	Avg. Metric	65.3	66.7	67.3	66.3
Secondary Actions	handshake	58.1	56.8	55.6	56.8
	hug	66.8	59.4	51.8	58.3
	read	70.7	55.2	50.6	59.0
	drink	69.5	47.7	37.8	49.0
	push-pull	74.0	65.5	63.1	68.1
	carry	76.5	78.0	78.3	77.4
	call	77.3	60.6	59.3	67.1
	Avg. Metric	70.4	60.4	56.6	62.7

**Table 3 sensors-22-07020-t003:** Comparison among different YoloV5 architectures. The first main comparison is between YoloV5-P5 and YoloV5-P6. The parameters are measured in millions (M), average precision is measured in %, training time is measured in hours, and model size is measured in MB. In the names of YoloV5, the subscripts “s”, “n”, “m”, “l”, and “x” refer to small, nano, medium, large, and extra-large network architectures.

Method	Layers	Parameters	GFlops	mAP	mAP@0.5	Recall	F-Score	Time	Size
YoloV5s	213	7.04	15.9	64.6	62.9	65.8	65.2	7.1	14.4
YoloV5x	444	86.2	204.2	74.5	49.7	51.9	61.2	32.8	173.2
YoloV5n	213	1.77	4.2	65.2	59.0	59.8	62.4	5.6	3.9
YoloV5m	290	20.9	48.1	72.3	67.8	67.3	69.7	12.7	71.2
YoloV5l	367	46.2	108	74.2	68.6	67.0	70.4	17.2	92.9
YoloV5s6	280	12.3	16.3	68.7	63.4	61.4	64.8	7.3	25.1
YoloV5x6	574	140.1	208.3	**75.4**	68.9	67.4	71.0	35.4	281.1
YoloV5n6	280	3.11	4.3	61.5	56.7	55.0	58.1	5.7	6.6
YoloV5m6	378	35.3	49.1	70.5	64.7	63.3	66.7	12.2	71.2
YoloV5l6	476	76.2	110.2	73.5	68.0	65.8	69.4	18	153.2

**Table 4 sensors-22-07020-t004:** Comparison of our proposed method with other state-of-the-art methods for the Okutama-Action dataset. We report the YoloV5x6 as the best performing Yolo architecture. The performance is measured in mean average precision (%) for 50% IoU. PA stands for primary actions, while SA stands for secondary actions. FFNN represents fully-connected feed-forward neural networks, while GB denotes gradient boosting.

Method	Image/Video	Architecture	mAP_0.5_	PA	SA
SSD512 [42]	Image	FFNN	15.4	-	-
SSD960 [42]	Image	FFNN	18.8	-	-
SSD+CNN [69]	Image	FFNN	28.3	-	-
POINet-Benchmark [39]	Video	LSTM	-	38.8	44.6
POINet-SFT [39]	Video	CNN–LSTM	-	26.5	17.9
Proposed Method	Image	YoloV5+GB	**75.4**	**71.8**	**77.9**

## Data Availability

Not applicable.

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
