# Peer review of "Detecting Human Actions in Drone Images Using YoloV5 and Stochastic Gradient Boosting"

_sensors, 2022, doi:10.3390/s22187020_

Round 1
Reviewer 1 Report
The paper is interesting, and some concerns should be addressed before publication:
In lines 25,26 and subsequent introduction, please explain each of them and avoid a bunch of references: “The recognition of human action is a well-studied problem [2], [3], [4], [5], [6], [7], but action detection in images is a less explored domain [9], [10], [11], [12], [13].”
The reference style should be based on MDPI reference style. For example, “[18] applies the Yolo architecture to small object”; [36] recorded a slow and low-altitude (about 10ft) UAV video dataset for detecting different gestures in aerial videos. This is not acceptable (Reference should be preceded by author name etc.).
The introduction and literature sections are very broad and do not discuss problems and challenges. The literature should demonstrate what were the problems in the existing techniques and how the proposed method has solved those problems.
The figures’ caption should be based on MDPI style (caption should be below the figures).
There are different Machine learning classifiers. The authors could implement a few other ML methods and compare their performance to justify that the GB is the best choice.
The conclusion should be more comprehensive by highlighting what has been done, what performance was achieved, what are the challenges and so on.
Reviewer 2 Report
This work uses a two stage architecture for detecting human actions from drone video footage.
The paper looks into different architectures of YOLOv5 to study performance and complexity yet the authors opt for an offline version. Having offline detection restricts use, therefore a clear indication on how to have a solution that is online is required.
It is not clear why the YoloV5 is the optimal choice because it offers various configurations. This argument does not seem to hold and there must be a strong technical argument to claim that this is the "optimal" choice.
There is no indication on how the parameters are selected. Information should be included on how these parameters were selected.
Future work section should be expanded.
Missing references - Human Detection in Drone Images Using YOLO for Search-and-Rescue Operations by Sergio Caputo, Giovanna Castellano, Francesco Greco, Corrado Mencar, Niccolò Petti & Gennaro Vessio
Improved YOLOv5: Efficient Object Detection Using Drone Images under Various Conditions by Hyun-Ki Jung and Gi-Sang Choi
Some typos:
Page 4 - "which includes R-CNN series [58], [59], [60]." -> "which include the R-CNN series [58], [59], [60]."
Page 5 - "decreases as it is increased the number of boosting rounds." -> "decreases as the number of boosting rounds is increased."
"The Gradient-boosting algorithm is explained here 1." where is 1?
Page 6 - In algorithm 1, should in the last equation gamma be gamma subscript m?
Page 11 - "[41] exercised SSD for detecting actions" -> "[41] used SSD for detecting actions"
Round 2
Reviewer 1 Report
The authors have addressed all the comments appropriately.
The conclusion can be further improved by explaining how the findings achieved better results instead of mentioning " we achieved better results for action detection than other methods in the literature that include explicit temporal information e.g. LSTM or 3D-CNN."(line 342-344).
Similarly, the last sentence of the abstract can be improved by explaining how better results were achieved.
Minor English problems are required to check before publication.
Author Response
Revision Notes for MDPI sensors-1880531
We would like to express our sincere appreciation to the editors and reviewers for their constructive comments on this submission. We have carefully followed all the comments given by the reviewers, and have made every possible effort to resolve every issue raised by the reviewers. We provide below detailed explanations of the changes that we have made in response to the reviewers’ comments.
To Reviewer #1:
Q1. The conclusion can be further improved by explaining how the findings achieved better results instead of mentioning " we achieved better results for action detection than other methods in the literature that include explicit temporal information e.g. LSTM or 3D-CNN."(line 342-344).
Author’s reply:
In the revised manuscript, we have better explained and rephrased this sentence.
In Section: Conclusion
“Since YoloV5 is efficient for detecting multiple objects in an image and Okutama-Action dataset contains concurrent actions in a frame, therefore, our proposed method achieved better results for single-image action detection than other methods in the literature that may include explicit temporal information e.g. LSTM or 3D-CNN.”
Q2. Similarly, the last sentence of the abstract can be improved by explaining how better results were achieved.
Author’s reply:
In the revised manuscript, we have better explained this sentence.
In Section 1: Introduction
“Our approach outperformed previous architectures applied to the Okutama-Dataset, which differed by their object identification and classification pipeline: we hypothesize that this is a consequence of both Yolov5 performance and the overall adequacy of our pipeline to the specificities of the Okutama-dataset in terms of bias-variance tradeoff”
Q3. Minor English problems are required to check before publication.
Author’s reply:
In the revised manuscript, we have checked for any possible English grammar and typos.